# The Impact of Healthcare Reforms on the Epidemiology Workforce in Kazakhstan: An Interrupted Time Series Analysis with Predictive Modeling of Nationwide Data Sources from 1998 to 2022

**DOI:** 10.3390/healthcare13020170

**Published:** 2025-01-16

**Authors:** Togzhan Akpanova, Tolebay Rakhypbekov, Yuliya Semenova, Akmaral Mussakhanova, Assiya Turgambayeva, Marina Zhanaliyeva, Ruslan Zharilkassimov, Sergey Kim, Aigerim Alzhanova, Raushan Sekenova, Marzhan Dauletyarova

**Affiliations:** 1Department of Science and Human Resources of the Ministry of Health of the Republic of Kazakhstan, Astana 010000, Kazakhstan; shalaganovatogzhan@gmail.com; 2Chair of Public Health and Management, Astana Medical University NJSC, Astana 010000, Kazakhstan; makmaral1@gmail.com (A.M.); assiya739@gmail.com (A.T.); zhanalieva.m@amu.kz (M.Z.); rzharilkassimov@gmail.com (R.Z.); sergeykimast@gmail.com (S.K.); alzhanovaaaak@gmail.com (A.A.); sekenova.r@amu.kz (R.S.); 3National Association «Primary Health Care», Astana 010000, Kazakhstan; tolebay52@inbox.ru; 4Department of Surgery, Nazarbayev University School of Medicine, Astana 010000, Kazakhstan; yuliya.semenova@nu.edu.kz; 5LLC «Next Event Group», Astana 010000, Kazakhstan

**Keywords:** epidemiology doctors, epidemiology nurses, infectious morbidity, interrupted time series analysis, Kazakhstan, sanitary-epidemiological facilities

## Abstract

Background: Following its independence, Kazakhstan implemented several reforms, including the adoption of the Entrepreneurial Code in 2008. This study aims to evaluate the impact of these reforms on the number and per capita rates of epidemiologists, nurse epidemiologists, epidemiological surveillance centers, and infectious morbidity from 1998 to 2022. Such an evaluation is critical for informing policy decisions regarding the future of epidemiological services in Kazakhstan. Methods: An interrupted time series analysis using a best-fit epidemiological model was conducted to assess the impact of key interventions—specifically, the adoption of the Entrepreneurial Code of the Republic of Kazakhstan and subsequent legislation—on the number and per capita rates of epidemiologists, nurse epidemiologists, and epidemiological surveillance facilities with infectious morbidity across the country. Results: Infectious morbidity per million individuals ranged from 4698.14 to 2263.79, with a consistent downward trend observed throughout the study period. Over the study period, the per capita rates of urban epidemiologists exhibited a downward trend, whereas the rates of rural epidemiologists showed an upward trajectory. The per capita rate of epidemiological surveillance centers declined from 26.89 to 15.24 over the study period. Substantial disparities were observed between urban and rural areas, with the epidemiology workforce in urban settings being 3–4 times larger than that in rural areas. Conclusions: This evaluation is important for informing policy decisions regarding the future of epidemiological surveillance services in Kazakhstan.

## 1. Introduction

Epidemiology is a key component of public health, providing scientific approaches for monitoring, preventing, and responding to population health threats. Recent studies, such as the Assessment of Epidemiology Capacity in State Health Departments [1], highlight the importance of systematically assessing epidemiological capacity, especially in the face of the increasing complexity of global challenges. Similarly, the article “One Field Epidemiologist per 200,000 Population: Lessons Learned from Implementing a Global Public Health Workforce Target [2]” demonstrates that the availability of a sufficient number of qualified specialists plays a critical role in increasing the resilience of health systems.

In Kazakhstan, health reforms implemented since the late 1990s have significantly changed the structure and functions of the epidemiological service. These changes include decentralization of management, the introduction of new financing methods, and integration with international standards, which inevitably affect the system’s ability to respond effectively to epidemiological challenges. However, the impact of these reforms on epidemiology remains poorly understood, making it difficult to assess their success and identify areas for further improvement.

The public health systems of post-Soviet states originated from the epidemiological surveillance service that dominated public health practices throughout the former Soviet Union. This service was organized hierarchically, from the top down, led by the Country Health Inspector, who also served as the Deputy Minister of Health. It was represented at all administrative levels, with divisions at the national, regional, city, and district levels. The epidemiological surveillance service also includes several central research centers and maintains a network of laboratories [3].

Starting from a similar point of departure upon gaining independence, post-Soviet countries embarked on various types of reforms. Some of them (including Armenia, Belarus, the Russian Federation, and Ukraine) largely retained the epidemiological surveillance system inherited from the Soviet era. Other countries (including Kazakhstan, Kyrgyzstan, Tajikistan, and Uzbekistan) have built additional structures [4]. Moreover, some countries (notably Georgia and the Republic of Moldova) abandoned Soviet-style epidemiological surveillance and created new public health infrastructures [5].

Armenia, Belarus, the Russian Federation, and Ukraine have focused their reform efforts on the traditional functions of epidemiological surveillance, continuing to prioritize infectious disease control and sanitary inspection. Azerbaijan, Kazakhstan, Kyrgyzstan, Tajikistan, and Uzbekistan have taken further steps by creating new structures to complement their epidemiological surveillance [6]. In Azerbaijan, epidemiological surveillance remains focused on infectious disease control and the oversight of immunization programs [7].

In Kazakhstan, during the 1990s, all local epidemiology surveillance centers were transferred to the control of local executive bodies, leading to ambiguity in institutional roles and responsibilities, weak governance, and a lack of coordination. However, from 2007 to 2008, the epidemiological surveillance system was restored to its previous vertical structure [8].

Kazakhstan also implemented a new vertical structure for healthy lifestyle centers, complementing the traditional public health model. These centers are governed by the National Center for Healthy Lifestyles [9,10].

In October 2015, the Entrepreneurial Code was adopted, which limited inspections of business entities, thereby impacting the functions of epidemiological surveillance. The introduction of the Entrepreneurial Code of the Republic of Kazakhstan in 2015 significantly affected sanitary and epidemiological services. The reduction in the number of inspections and the transition to a risk-based approach changed its functions, reducing the workload of employees. Part of the service’s tasks were transferred to local authorities and the private sector, which led to a reduction in the number of employees. Moreover, the role of sanitary-epidemiological services was strengthened in the areas of prevention and counseling.

In 2017, under this Code, the Ministry of National Economy issued subordinate regulations that significantly restricted the service’s functions and reduced state orders for workforce training [11,12].

In 2017, the epidemiological surveillance service was incorporated into the Committee for Public Health Protection within the Ministry of Health of the Republic of Kazakhstan. In 2019, it was subsequently transferred to the newly formed Committee for Quality and Safety Control of Goods and Services [13].

To date, no studies have investigated the impact of these reforms on the number and density of epidemiological surveillance facilities and epidemiologists from the dissolution of the Soviet Union to the present day in Kazakhstan. This study aims to assess the impact of these reforms on the number and per capita rates of epidemiologists, nurse epidemiologists, and epidemiological surveillance centers over a period spanning more than two decades (from 1998 to 2022). Additionally, the aim of this study was to examine the relationship between infectious morbidity rates and the state of the country’s epidemiological services. Special emphasis was placed on the impact of the adoption of the Entrepreneurial Code. This evaluation is important for informing policy decisions regarding the future of sanitary services in Kazakhstan.

## 2. Materials and Methods

### 2.1. Study Design and Data Sources

This study employed a retrospective design and analyzed nationwide data spanning 1998–2022. Various data sources have been utilized, with the primary source being official statistical reports published annually by the Ministry of Health (MoH) [14,15] (see Appendix A). These reports include a subsection on the healthcare network and personnel and population health indicators (total morbidity of the population of the Republic of Kazakhstan by classes of diseases) disaggregated by region (oblast) and location (urban versus rural). The earliest report dates back to 1998, and the most recent available report is from 2022. Further details on the use of these data sources for healthcare system analysis are available in previous studies [16,17]. To calculate per capita rates, midyear population data were obtained from the Bureau of National Statistics through their annual statistical reports [17].

### 2.2. Study Units

The study units included the numbers of epidemiologists, nurse epidemiologists, and epidemiological surveillance centers. Data on the total number of epidemiologists and nurse epidemiologists in Kazakhstan were available and extracted for the period from 1998 to 2022. In addition, data on the number of epidemiologists and nurse epidemiologists practicing in urban and rural areas were available and extracted for the period from 2000 to 2022. Information on the total number of epidemiological surveillance centers in Kazakhstan was available and extracted for the period from 2003 to 2022.

### 2.3. Study Formulas

The per capita number of epidemiologists, nurse epidemiologists, and epidemiological surveillance centers was calculated per million population (PMP) via the following formulas:Per capita rates of epidemiologists = number of epidemiologists/midyear population × 1,000,000.Per capita rates of nurse epidemiologists = number of nurse epidemiologists/midyear number of population × 1,000,000.Per capita rates of epidemiological surveillance centers = number of epidemiological surveillance centers/midyear population × 1,000,000.

The mean change was derived via the following formula:Mean change = mean value after the intervention − mean value before the intervention.

The mean change related to the preintervention period was expressed as a percentage and computed via the following formula:The number of infectious morbidities per epidemiologist = the annual number of infectious morbidities/the total number of epidemiologists in the same year.Mean change related to the preintervention period = (mean change/mean before intervention) × 100.

### 2.4. Definitions Used in This Study

An epidemiologist is a specialist with higher medical or biological education and is trained in epidemiology. An epidemiologist studies the patterns of infectious and noninfectious diseases, analyzes the epidemiological situation, and develops and implements disease prevention and control measures.

Centers for epidemiological surveillance are specialized institutions or units of sanitary-epidemiological services that monitor infectious and noninfectious disease incidence, analyze epidemiological data, and develop measures to prevent and control disease outbreaks. These centers also conduct educational and methodological work with medical institutions.

Infectious morbidity—Infectious disease incidence is a measure of the prevalence of infectious diseases in a given population over a given period of time. It is used to analyze the epidemiological situation and to plan preventive measures. This term encompasses data on the number of cases, incidence by age and occupational group, and pattern of spread of infection (outbreak or epidemic).

A nurse epidemiologist is a medical worker with specialized secondary medical education in the field of epidemiological surveillance. She is engaged in collecting and analyzing data on disease incidence, monitoring the implementation of sanitary and hygienic norms, and organizing preventive measures.

### 2.5. Data Analysis

The Statistical Package for Social Sciences (IBM SPSS Statistics) version 24.0 was employed to conduct various time series tests. The average annual change, along with the 95% confidence interval (95% CI), was calculated to assess changes over the study period. Interrupted time series analysis (ITSA) was utilized to evaluate the impact of the adoption of the Entrepreneurial Code of the Republic of Kazakhstan and subsidiary legislation in 2017 on the numerical counts and per capita rates of epidemiologists, nurses, epidemiologists, epidemiological surveillance centers, and infectious morbidity. The Expert Modeler function of SPSS was used to identify the best-fit epidemiological model. The variables corresponding to the interventions were subsequently categorized as ‘events’, and the model was subsequently adjusted on the basis of parameters identified by the best-fit model. The percentage point change (PPC) value was extracted to estimate the impact of each intervention.

To project the future numbers of epidemiologists, nurse epidemiologists, and epidemiological surveillance centers, the Expert Modeler function of SPSS was employed to identify the best-fit forecasting model, and corresponding values until 2030 were extrapolated [16]. The Expert Modeler function of SPSS was employed to identify the best-fit forecasting model, and the corresponding values until 2030 were extrapolated. All tests were considered significant at *p* = 0.05.

For a visual representation of the mean numbers and PMP rates of epidemiologists, nurses, epidemiologists, and epidemiological surveillance centers, maps of Kazakhstan were generated via QGIS 3.26 “Buenos Aires” software.

### 2.6. Ethics Statement

Prior to the beginning of the study, the Ethics Committee of Astana Medical University reviewed the study protocol and waived the requirement for informed consent (Minutes of meeting # 2 dated, 2021).

## 3. Results

As depicted in Figure 1, since 2017, the number of epidemiological surveillance centers has decreased (A). The total number of epidemiologists (B) and urban epidemiologists (C) experienced a temporary increase from 2017, followed by a sharp decline starting in 2019. The number of rural epidemiologists (D) has been growing since 2017, whereas the number of nurse epidemiologists has been decreasing (E).

Throughout the period spanning from 1998 to 2022, the total number of epidemiologists fluctuated within a range of 3164 (in 1998)–4287 (in 2016), whereas the number of urban epidemiologists varied between 2541 (in 2000) and 3574 (in 2016). Similarly, the number of rural epidemiologists ranged from 555 (in 2000) to 713 (in 2016). When the means before and after the intervention periods were compared, a negative difference was observed for the total number of epidemiologists, the number of urban epidemiologists, the number of rural epidemiologists, and the total number of epidemiological surveillance centers, as well as their respective per capita rates (Table 1).

The analysis conducted via the ITSA, which utilized a best-fit epidemiological model, indicated that the adoption of the Entrepreneurial Code and subsidiary legislation in 2017 had a statistically significant negative effect on both the numerical count and per capita rates of epidemiological surveillance centers and total epidemiologists. Specifically, there was a substantial decrease of 201% in the numerical count and PMP rates of rural epidemiologists. However, this intervention did not have a negative effect on the number and PMP rates of total and urban epidemiologists or nurse epidemiologists (Table 2).

Between 1998 and 2022, the number of infectious morbidities in Kazakhstan generally decreased. The average rate of change during the preintervention period was negative, with a decline of −12.50% following the introduction of the impact of the adoption of the Entrepreneurial Code and subsidiary legislation in 2017 (see Table 3).

Kazakhstan is divided into 16 administrative units: 14 regions and 2 cities of republican significance (Astana and Almaty). The regions are located throughout the country: North Kazakhstan, Kostanay—in the north; Akmola, Karaganda—in the central part; Aktobe, Atyrau, West Kazakhstan, and Mangystau—in the west; East Kazakhstan, Pavlodar—in the east; Almaty—in the southeast; and Zhambyl, Kyzylorda, and Turkestan—in the north and in the south. Figure 2 illustrates the distribution of the epidemiology workforce and epidemiological surveillance centers across the regions of Kazakhstan. The dark red shading represents the highest incidence rates, whereas the light gray shading represents the lowest incidence rates. This figure highlights the differences between the periods before (1998–2016) and after (2017–2022) the introduction of the Entrepreneurial Code. Specifically, from 1998 to 2016, the South Kazakhstan region had the lowest percentage of epidemiologists in the country (145.35 per 100,000 people), whereas the Karaganda region had the highest percentage (372.20 per 100,000 people). However, during the period from 2017 to 2022, the regional patterns changed. The Mangystau region had the highest rate of epidemiologists (299.73 per 100,000 people), whereas the lowest rate was observed in the North Kazakhstan region (81.4 per 100,000 people) (Figure 2A,B).

In terms of the rates of nurse epidemiologists from 2000 to 2016, the lowest rate was observed in the Almaty region (164.58 per 100,000 people), whereas the highest rate was in the West Kazakhstan region (340.33 per 100,000 people). From 2017 to 2022, the lowest rate was in South Kazakhstan (80.67 per 100,000 people), and the highest rate was in the Kyzylorda region (325.13 per 100,000 people), as shown in Figure 2C,D. The number of epidemiological surveillance centers prior to the adoption of the Entrepreneurial Code and subsidiary legislation in 2017 was the lowest in the South Kazakhstan region (14.72 per 100,000 population) and the highest in the Kostanay region (62.9 per 100,000 population). After the adoption of the Entrepreneurial Code (the period spanning from 2017 to 2022), no epidemiological surveillance centers existed in the Almaty region, despite this region previously having an average rate within the Republic (23.16 per 100,000 people). The highest number of epidemiological surveillance centers during this period was in the Kostanay region (44.29 per 100,000 population) (Figure 2E,F).

Projections for the total number of epidemiologists up to 2030 indicate an increasing trend, whereas the number of epidemiological surveillance centers and nurse epidemiologists is expected to decline. However, the number of infectious morbidities in Kazakhstan is likely to have tended to decrease. Similarly, the numbers of both urban and rural epidemiologists are projected to increase, reflecting the overall growth in the total number of epidemiologists (Figure 3).

Table 4 shows that by 2030, Kazakhstan is projected to have 256.1 epidemiological surveillance centers (95% CI: 256.1; 470.4), 3777 epidemiologists (95% CI: 2614; 3941), and 2168 nurse epidemiologists (95% CI: 233; 4102). Infectious morbidity is expected to be 356,237.7 (95% CI: 569,350.1; 143,125.3). Moreover, the number of urban epidemiologists is expected to be 2975 (95% CI: 1905; 4044), whereas the number of rural epidemiologists is projected to be 841 (95% CI: 225; 1458). Notably, the rates of rural epidemiologists (t = 38.929, *p* < 0.001) and infectious morbidity (t = −2.29, *p* = 0.031) were significantly affected in this study.

## 4. Discussion

Overall, the per capita rates of epidemiological surveillance centers ranged from 26.89 to 15.24, with a declining trend observed during the study period (1998–2022). The adoption of the Entrepreneurial Code and subsidiary legislation had a significantly negative effect on both the number and per capita rates of rural epidemiologists, whereas its impact on urban epidemiologists was minimal. To gain deeper insights into this issue, these findings should be considered alongside other studies addressing similar concerns.

The authors of Becerra AZ et al. emphasized the importance of diversity and inclusion in the epidemiology workforce [18]. According to studies by the group of authors Williams SG, Fontaine RE, Turcios Ruiz RM, Walke H, Ijaz K, and Baggett HC, in the USA, there is one field epidemiologist per 200,000 people [2]. However, if we compare this standard with that of Kazakhstan, we need to pay attention to the fact that historically, a high density of medical personnel, including doctors and nurses, compared with other countries, has been dictated by the Semashko system, which Kazakhstan has adhered to for many years [4].

Our study shows the direct impact of the Entrepreneurial Code on the staffing structure: reducing the number of health workers was one of the optimization measures aimed at improving the efficiency of the health care system. This reduction, in turn, may have had long-term consequences for the sustainability of staffing and the quality of health services.

Overall, the per capita rates of epidemiological surveillance centers ranged from 26.89 to 15.24, with a declining trend observed during the study period (1998–2022). The adoption of the Entrepreneurial Code and subsidiary legislation had a significantly negative effect on both the number and per capita rates of rural epidemiologists, whereas its impact on urban epidemiologists was minimal. To gain deeper insights into this issue, these findings should be considered alongside other studies addressing similar concerns.

Upon gaining independence, post-Soviet countries undertook various reforms: Armenia, Belarus, the Russian Federation, and Ukraine largely retained the epidemiological surveillance systems inherited from the Soviet era; Kazakhstan, Kyrgyzstan, Tajikistan, and Uzbekistan developed additional structures; and Georgia and the Republic of Moldova abandoned the Soviet epidemiological surveillance system and established new public health infrastructures [4]. Since the 1990s, the number of healthcare workers specializing in epidemiology and public health in the Russian Federation has significantly decreased by nearly half (from 27,800 to 14,000 individuals) [19]. However, in recent years, the epidemiology workforce has increased, with approximately 14,000 epidemiologists practicing in Russia in 2021, slightly more than in 2018. This growth is likely attributable to the heightened role of epidemiologists in combating the coronavirus disease 2019 (COVID-19) pandemic [19].

In Uzbekistan, the epidemiological surveillance service is the authorized state body responsible for implementing a unified state policy in infection control and public health. This service is part of the MoH, with its head also serving as the Deputy Minister of Health and the Chief State Sanitary Doctor of the Republic of Uzbekistan [20]. Until 2020, epidemiologists were not in high demand in Uzbekistan, ranking sixth in number behind general practitioners, surgeons, dentists, pediatricians, and obstetrician-gynecologists. Over the past decade, the number of epidemiologists has fluctuated between 4200 and 4600, despite the country’s growing population. For the last 10 years, Uzbekistan has consistently maintained a ratio of 1 to 1.5 epidemiologists per 1000 people [21].

Over the past 20 years, following various reorganizations of the epidemiological surveillance service in the Kyrgyz Republic, this public health service is currently represented by six national-level organizations: the Department for Disease Prevention and State Sanitary-Epidemiological Surveillance, the Republican Immunoprophylaxis Center, the Republican Center for Quarantine and Especially Dangerous Infections, the Republican AIDS Center, the Republican Center for Health Promotion, and the Scientific and Production Association “Preventive Medicine”. Additionally, there are 19 interdistrict and 2 district centers for epidemiological surveillance: the Bishkek City Center for State Sanitary-Epidemiological Surveillance, Centers for State Sanitary-Epidemiological Surveillance in transport, three anti-plague departments (located in Osh, At-Bashi, and Karakol), Bishkek city and southern branches of the Health Promotion Center, and 8 regional AIDS control centers [22]. From 1997 to 2023, the number of epidemiologists practicing in the Kyrgyz Republic decreased by 32%, totaling 521 individuals [23].

A study by Kairzhan et al. reported that in 2020, during the COVID-19 pandemic, the MoH of the Republic of Kazakhstan faced a shortage of more than 800 epidemiologists and nurse epidemiologists, which significantly impacted the fight against COVID-19 [24]. In comparison, the present study revealed that the PMP rates of epidemiologists and nurse epidemiologists ranged from 209.93 to 184.69 and from 208.14 to 122.82, respectively, with a decreasing trend observed throughout the study period. However, notable differences were observed between urban and rural areas, with the workforce of epidemiologists in cities being 3–4 times larger than that in rural areas. To address the situation with epidemiologists in the Republic of Kazakhstan, active efforts have been underway since May 2023. According to Order No. 86 issued by the Acting Minister of Health of the Republic of Kazakhstan on 24 May 2023, an educational program for continuous integrated medical education in the field of epidemiological surveillance is being developed. Upon completion of this program, graduates will be qualified as medical hygienists and epidemiologists [25]. In 2023, the state educational order allocated 100 grants for this training program nationwide, with an additional 100 grants designated for 2024 [26].

To address the shortage of health workers in rural areas of Kazakhstan, it is necessary to introduce a package of social benefits aimed at attracting them. Measures such as the provision of housing, financial support, and additional benefits can significantly increase the motivation of specialists to work in rural areas. Studies show that such an approach helps increase the number of medical personnel in rural areas, reduces staff shortages, and helps to eliminate the imbalance in access to medical services between urban and rural areas [16].

According to research findings, the incidence of infectious diseases per million people ranged from 4698.14 to 2263.79, with a consistent declining trend observed throughout the study period. However, the focus has shifted from infectious disease incidence to antimicrobial resistance (AMR). The World Health Organization has identified AMR as one of the most critical global public health challenges, which is projected to result in significant additional healthcare costs, hospitalizations, and mortality, potentially affecting millions of people by 2050 [27]. Without preventive measures, it is estimated that by 2050, AMR could become the leading cause of death worldwide [28]. In this context, studies by Mounesan et al. suggest that the role of epidemiologists is expanding to include tasks such as identifying issues at various levels, analyzing trends, modeling resistance pathways, and educating target groups [29]. This trend is particularly relevant for Kazakhstan, where clearly defining the role of epidemiologists in collaboration with other stakeholders to accelerate preventive and control measures aimed at addressing the growing threat of AMR is crucial.

The primary strength of this study is that it represents the first investigation into the impact of reforms on the number and density of the epidemiology workforce in Kazakhstan over an extended period. Additionally, this is the first report to provide national and regional numbers and rates for epidemiology workers, epidemiological surveillance centers, and infectious morbidity. A key limitation of this study is the aggregated nature of the data, which restricts the level of detail in the analyses that can be performed. Future research should investigate the impact of other reforms on the numbers and per capita indicators of additional public health subspecialties in Kazakhstan.

## 5. Conclusions

The reforms implemented in the Republic of Kazakhstan, particularly the adoption of the Entrepreneurial Code and its subsidiary legislation, had a significant negative impact on the epidemiology workforce and epidemiological surveillance centers, as evidenced by the decreasing trend observed from 1998 to 2022. Significant disparities also emerged between urban and rural areas. In addition, the COVID-19 pandemic has presented shortcomings in the field of epidemiological surveillance, highlighting the need for systemic improvements. This study will be crucial for informing policy decisions and enhancing epidemiological surveillance services across the country and its regions.

## Figures and Tables

**Figure 1 healthcare-13-00170-f001:**
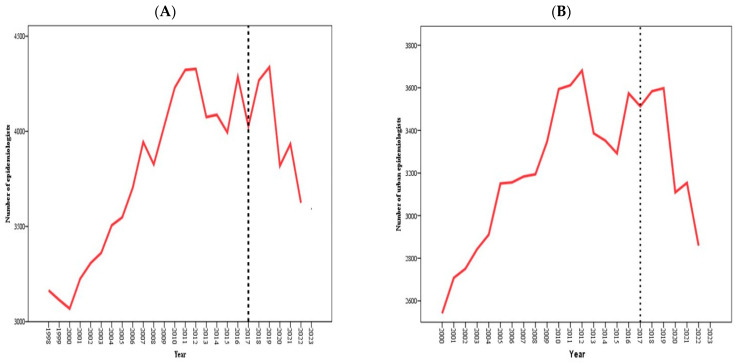
Numbers of total epidemiologists (**A**), urban epidemiologists (**B**), rural epidemiologists (**C**), nurse epidemiologists (**D**), and epidemiological surveillance centers (**E**), 1998–2022. The gray dotted lines represent one intervention: the introduction of the Entrepreneurial Code of the Republic of Kazakhstan and subsidiary bylaws in 2017.

**Figure 2 healthcare-13-00170-f002:**
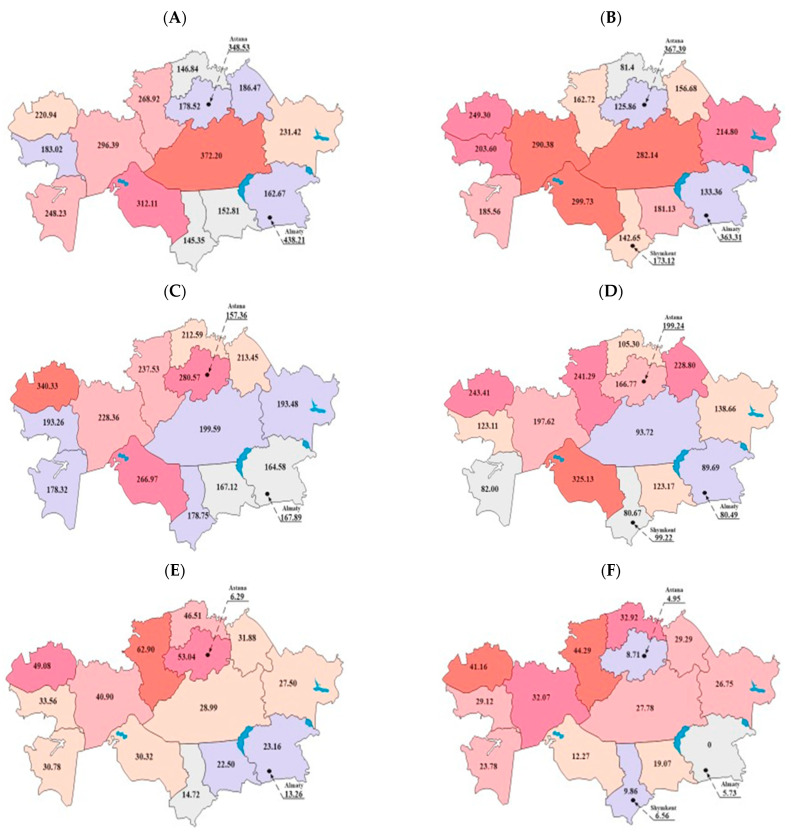
Mean per million population rates of epidemiologists during the periods 1998–2016 (**A**) and 2017–2022 (**B**), nurse epidemiologists during the periods 2000–2016 (**C**) and 2017–2022 (**D**), and epidemiological surveillance centers during the periods 2003–2016 (**E**) and 2017–2022 (**F**) across different regions of Kazakhstan.

**Figure 3 healthcare-13-00170-f003:**
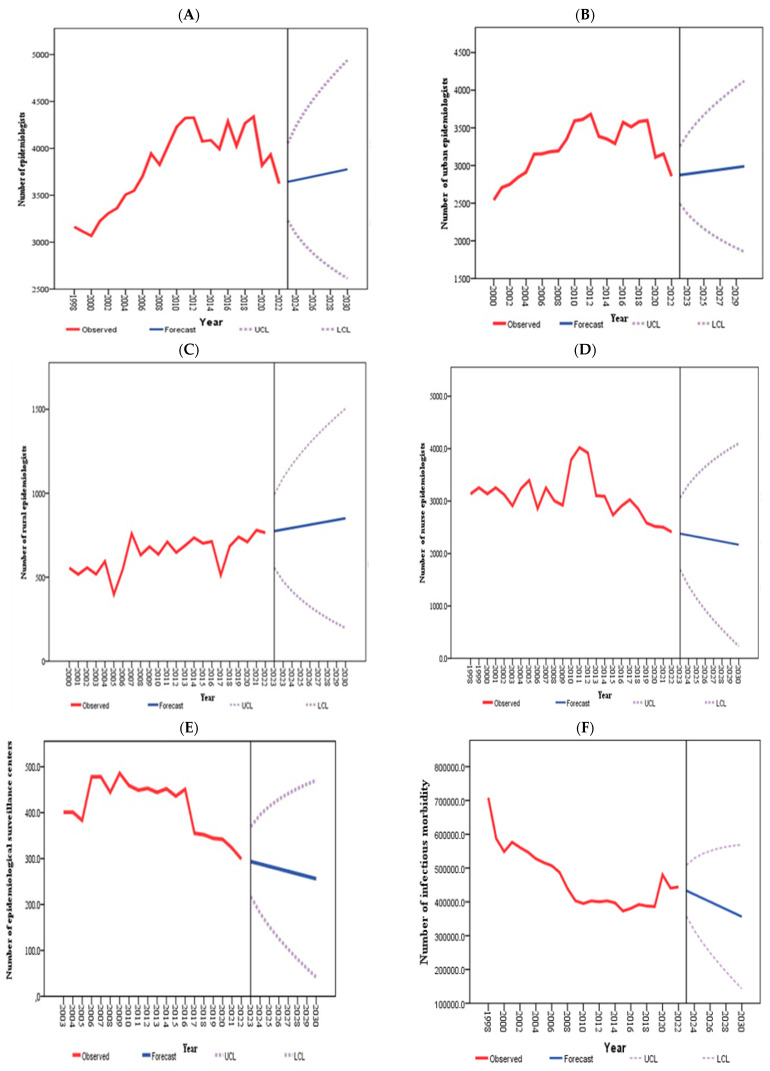
The observed and projected numbers of epidemiologists (**A**), urban epidemiologists (**B**), rural epidemiologists (**C**), nurse epidemiologists (**D**), epidemiological surveillance centers (**E**), and infectious morbidities (**F**).

**Table 1 healthcare-13-00170-t001:** Descriptive statistics of the number of epidemiological surveillance centers, epidemiologists, and nurse epidemiologists and the impact of the adoption of the Entrepreneurial Code and subsidiary legislation in 2017.

		Mean Before Intervention	Mean After Intervention	Mean Change
the adoption of the Entrepreneurial Code of the Republic of Kazakhstan and subsidiary legislation in 2017				
Number	Total epidemiologists	3747.16	4001.33	258.17
	Urban epidemiologists	3192.76	3302.83	110.07
	Rural epidemiologists	623.12	698.5	75.38
	Total nurse Epidemiologists	3213.42	2648.83	−564.59
	Total epidemiological surveillance centers	443.93	335.83	−108.83
Per million population rates	Total epidemiologists	235.22	214.39	−20.83
	Urban epidemiologists	351.93	307.44	−44.48
	Rural epidemiologists	89.79	88.02	−1.76
	Nurse epidemiologists	203.09	142.03	−61.06
	Total epidemiological surveillance centers	27.42	17.99	−9.43

**Table 2 healthcare-13-00170-t002:** Interrupted time series analysis of changes in the numbers and rates of epidemiological surveillance centers, epidemiologists, and nurse epidemiologists: effects of the adoption of the Entrepreneurial Code and subsidiary legislation in 2017.

Model Component	Model	Stationary R Squared	Estimate (PPC *)	*p* Value
Total number of epidemiologists	Brown (0.1.0)	0.602	0.446	<0.001
Number of urban epidemiologists	Simple	−0.007	0.994	<0.001
Number of rural epidemiologists	ARIMA (0.1.0)	0.167	−201.0	0.048
Total number of nurse epidemiologists	ARIMA (0.1.0)	0.002	0.869	<0.001
Number of epidemiological surveillance centers	ARIMA (0.1.0)	0.370	−96.0	0.004
Rate of total epidemiologists	ARIMA (0.1.0)	˂0.001	−0.005	0.625
Rate of urban epidemiologists	Brown	0.446	0.573	<0.001
Rate of rural epidemiologists	Simple	−2.309	89.327	<0.001
Rate of nurse epidemiologists	Simple	−0.02	0.871	<0.001
Rate of epidemiological surveillance centers	ARIMA (0.1.0)	0.285	−5.665	0.007

* PPC—percentage point change.

**Table 3 healthcare-13-00170-t003:** Descriptive statistics of the number and rate of infectious morbidities: the impact of the adoption of the Entrepreneurial Code and subsidiary legislation in 2017.

Indicator	Mean Before Intervention	Mean After Intervention	Mean Change	Mean Change Related to Preintervention Period, %
the adoption of the Entrepreneurial Code of the Republic of Kazakhstan and subsidiary legislation in 2017				
Number of infectious morbidity	482,159.2	421,873	−60,286.2	−12.50
Infectious morbidity rate	3077.84	2254.22	−823.619	−26.76
Number of infectious morbidities per epidemiologist	128.82	105.43	−23.38	−18.15

**Table 4 healthcare-13-00170-t004:** The projected numbers and rates of epidemiologists, total epidemiologists, urban epidemiologists, rural epidemiologists, and nurse epidemiologists and infectious morbidity for the years 2026 and 2030, accompanied by 95% confidence intervals.

Number	Year		Model Parameters		
	2026 Rate (95% CI *)	2030 Rate (95% CI)	Type of Model	Alpha (Level)	
				T	*p* Value
Total epidemiologists	3701 (2878; 4523)	3777 (2614; 4941)	ARIMA (0.1.0)	0.472	0.641
Urban epidemiologists	2917 (2160; 3673)	2975 (1905; 4044)	ARIMA (0.1.0)	0.373	0.713
Rural epidemiologists	803 (367; 1239)	841 (225; 1458)	ARIMA (0.1.0)	0.427	0.673
Total nurse epidemiologists	2289 (921; 3657)	2168 (233; 4102)	ARIMA (0.1.0)	−0.449	0.658
Epidemiological surveillance centers	277.5 (126; 429.1)	256.1 (256.1; 470.4)	ARIMA (0.1.0)	−0.649	0.525
Infectious morbidity	400,218.8 (249,525.6; 550,912.1)	356,237.7 (569,350.1; 143,125.3)	ARIMA (0.1.0)	−1.479	0.153
Rate of total epidemiologists	181.8 (145.32; 224.91)	178.954 (129.948; 241.001)	ARIMA (0.1.0)	−0.495	0.625
Rate of urban epidemiologists	241.607 (162.393; 320.821)	231.957 (119.932; 343.982)	ARIMA (0.1.0)	−0.594	0.559
Rate of rural epidemiologists	89.327 (66.505; 112.149)	89.327 (66.505; 112.149)	ARIMA (0.0.0)	38.929	<0.001
Rate of nurse epidemiologists	114.905 (73.402; 172.378)	107.499 (56.329; 188.405)	ARIMA (0.1.0)	−1.044	0.308
Rate of epidemiological surveillance centers	12.783 (3.516; 22.051)	10.329 (−2.777; 23.435)	ARIMA (0.1.0)	−1.213	0.241
Rate of infectious morbidity	1858.1 (961.54; 2754.58)	1452.3 (184.47; 2720.21)	ARIMA (0.1.0)	−2.29	0.031

* 95% CI—95% confidence interval.

## Data Availability

The datasets used and/or analyzed during the current study are available from the corresponding author upon reasonable request.

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
