# Peer review of "The Impact of Healthcare Reforms on the Epidemiology Workforce in Kazakhstan: An Interrupted Time Series Analysis with Predictive Modeling of Nationwide Data Sources from 1998 to 2022"

_healthcare, 2025, doi:10.3390/healthcare13020170_

Round 1
Reviewer 1 Report
Comments and Suggestions for Authors
It is an interesting manuscript and worth documenting; however, there is room to improve, as indicated in the commented file.

Author Response
Thank you very much for taking the time to review our manuscript. We have carefully considered all your comments and made revisions accordingly, with the changes highlighted in yellow. Please refer to the manuscript file to view the changes made.
Comment 1: Previously, author aim to also investigate the infectious morbidity but no show here.
Response 1: Thank you for pointing this out. We agree with this comment. Therefore, we have adjusted this proposal and supplemented the methods as follows «Methods: An interrupted time series analysis using best-fit epidemiological model was conducted to assess the impact of key interventions—specifically, the adoption of the Entrepreneurial Code of the Republic of Kazakhstan and subsequent legislation—on the number and per capita rates of epidemiologists, nurse epidemiologists, and epidemiological surveillance facilities and infectious morbidity across the country».
Comment 2: I recommended author present the results with the same order throughout the manuscript as stated in the aims. This would be easy for reader to follow.
Response 2: Thank you for your careful study of the work and we have changed according to your recommendations and according to the objectives , so figure B-A, C-B,D-C,E-D,A-E (Figure1) and also in the tables.
Comment 3: Could author please mainly focus to emphasize the policy of 2017 (Entrepreneurial Code and subsidiary legislation) and describe the rationale of how the policy would impact all the interested outcome in this study? This would help convince readers.
Response 3: Agree. Accordingly, we have supplemented to emphasize this point. The introduction of the Entrepreneurial Code of the Republic of Kazakhstan in 2017 significantly affected the sanitary and epidemiological service. The reduction in the number of inspections and the transition to a risk-based approach changed its functions, reducing the workload of employees. Part of the service's tasks were transferred to local authorities and the private sector, which led to a reduction in the number of employees. At the same time, the role of the sanitary-epidemiological service was strengthened in the areas of prevention and counseling.
Comment 4: I recommended the author put some operation definition terms for epidemiologist, nurse epidemiologists, epidemiological surveillance centers and infectious morbidity. For example, epidemiologist refer to those attended / complete some course or something. This would be very useful
Response 4: Agree. We have, accordingly, modified section of materials and methods with this definition to emphasize this point.
Epidemiologist - an epidemiologist is a specialist with a higher medical or biological education, trained in epidemiology. An epidemiologist studies the patterns of infectious and non-infectious diseases, analyzes the epidemiological situation, and develops and implements disease prevention and control measures.
Centers for epidemiological surveillance are specialized institutions or units of the sanitary-epidemiological service that monitor infectious and non-infectious disease incidence, analyze epidemiological data and develop measures to prevent and control disease outbreaks. Such centers also conduct educational and methodological work with medical institutions.
Infectious morbidity - Infectious disease incidence is a measure of the prevalence of infectious diseases in a given population over a given period of time. It is used to analyze the epidemiological situation and to plan preventive measures. This term encompasses data on the number of cases, incidence by age and occupational groups, and the pattern of spread of infection (outbreak or epidemic).
Nurse-epidemiologist is a medical worker with specialized secondary medical education in the field of epidemiological surveillance. She is engaged in collecting and analyzing data on disease incidence, monitoring the implementation of sanitary and hygienic norms, and organizing preventive measures.
Comment 5: Where the number of infectious morbidity was extracted?
Response 5: The study design and data sources was added.
Comment 6: How about the number of infectious mordibity per epidemiologists?
Response 6: Data on epidemiologists were taken from official statistical reports published annually by the Ministry of Health (MoH) [12-13]. The formula for calculating the number of infectious morbidity cases per epidemiologist is presented in Study formulas (The number of infectious morbidity per epidemiologists= the annual number of infectious morbidity / the total number of epidemiologists in the same year).
Comment 7: Is the subsidiary legislation was limited in only 2017, otherwise the multiple intervention time point may be needed to introduced. /
Response 7: Only in 2017, changes in legislation were introduced with the adoption of the Entrepreneurial Code, which significantly affected the sanitary and epidemiological service. The reduction in the number of inspections and the transition to a risk-oriented approach changed its functions, reducing the workload of employees. Some of the service's tasks were transferred to local authorities and the private sector, which led to a reduction in the number of staff. There were no other interventions on the service during the period under study.
Comment 8: Please specify which indicator/index and criteria of selection that author use in this analysis.
Response 8: We use interrupted time series analysis (ITSA), which was utilized to evaluate the impact of interventions the adoption of the Entrepreneurial Code of the Republic of Kazakhstan and subsidiary legislation in 2017—on the numerical counts and per capita rates of epidemiologists, nurses, epidemiologists, epidemiological surveillance centers and and infectious morbidity. The Expert Modeler function of SPSS was used to identify the best-fit epidemiological model. The variables corresponding to the interventions were subsequently categorized as 'events', and the model was subsequently adjusted on the basis of parameters identified by the best-fit model.
Comment 9: Just Noted that I believe this showed not a good fit for ARIMA. not to be confident to claim the result.
Response 9: Thank you for pointing out the inconsistency. We have corrected it to “Simple” in Rate of rural epidemiologists.
Comment 10: Although author has discussed the trend of epidemiologists; however, I believe it is worth to discuss the number of epidemiologist per capita in a present is quite high compare other over some recommendation/estimated in some studies such as Arrazola, J. (2022). Assessment of Epidemiology Capacity in State Health Departments—United States, 2021. MMWR. Morbidity and Mortality Weekly Report, 71. https://doi.org/10.15585/mmwr.mm7113a2 Williams, S. G., Fontaine, R. E., Turcios Ruiz, R. M., Walke, H., Ijaz, K., & Baggett, H. C. (2020). One Field Epidemiologist per 200,000 Population: Lessons Learned from Implementing a Global Public Health Workforce Target. Health Security, 18(S1), S113–S118. https://doi.org/10.1089/hs.2019.0119 PS: I think discussion to give some view point on big difference number on different context is worth to be documented. Furthermore, beside the comparison, please also discuss deeply how the 2017 Entrepreneurial Code and subsidiary legislation impact all outcomes. Please discuss why nurse epidemiologist declined even during pandemic
Response 10: The discussion section was added.

Reviewer 2 Report
Comments and Suggestions for Authors
Abstract:
- Keywords: please sort alphabetically
Introduction:
- Please mention some similar studies in other countries and what part of this study is different from previous studies
Results
- Lines 194-195: table 1: please write the table neatly, adjust the spacing
- Lines 194-195: table 1: What is the meaning of "Mean change related to the preintervention period, %". Why are "Mean" and "%" in the same section?
- Lines 272-273: many empty spaces in the manuscript are not used; please adjust and type properly
- Lines 292-303: many empty spaces in the manuscript are not used. Please adjust and type properly
- Lines 318-319: table 4: please write the table neatly, adjust the spacing
- Lines 318-319: table 4 has not been clearly described in the results about the interpretation of "t" and 'p value"
Discussion
- Lines 321-326: There is no need to repeat to state the purpose of this study. The purpose of the study is sufficiently explained in the introduction
- In this section, the author only describes the study's findings and mentions the related history. Still, we have not seen the relationship between this study and the results of previous studies.
- Authors need to analyze the findings in this study to compare them with similar studies that have been conducted
Conclusion
- Please provide recommendations based on the findings in this study to address the "negative impacts" and "significant disparities between urban and rural areas."
References
- Reference number 1: is there no newer source? 1976 is too old to be used as a reference source
- Please correct the format for writing references number 17-21
Comments on the Quality of English LanguageIn some sections, the English writing needs improvement
Author Response
Response to Reviewer 2 Comments
Thank you very much for taking the time to review our manuscript. We have carefully considered all your comments and made revisions accordingly, with the changes highlighted in yellow. Please refer to the manuscript file to view the changes made.
Comment 1: Keywords: please sort alphabetically
Response 1: Thank you for pointing this out. We agree with this comment and sort keywords alphabetically.
Comment 2: Introduction: Please mention some similar studies in other countries and what part of this study is different from previous studies
Response 2: Thank you for your recommendation, the introducation section was added.
Comment 3: Results Lines 194-195: table 1: please write the table neatly, adjust the spacing
Response 3: Agree. We have, modified table 1.
Comment 4: Lines 194-195: table 1: What is the meaning of "Mean change related to the preintervention period, %". Why are "Mean" and "%" in the same section?
Response 4: Thank you, we corrected this issue.
Comment 5: Lines 272-273: many empty spaces in the manuscript are not used; please adjust and type properly
Response 5: Thank you for the recommendation, we have corrected this issue.
Comment 6: Lines 292-303: many empty spaces in the manuscript are not used. Please adjust and type properly
Response 6: Thank you for the recommendation, we have corrected this issue.
Comment 7: Lines 318-319: table 4: please write the table neatly, adjust the spacing
Response 7: Agree. We have, modified table 4.
Comment 8: Lines 318-319: table 4 has not been clearly described in the results about the interpretation of "t" and 'p value"
Response 8: Тhank for this recommendation, this information was added to results.
Comment 9: Lines 321-326: There is no need to repeat to state the purpose of this study. The purpose of the study is sufficiently explained in the introduction
Response 9: Done.
Comment 10: Authors need to analyze the findings in this study to compare them with similar studies that have been conducted
Response 10: The discussion was added.
Comment 11: Please provide recommendations based on the findings in this study to address the "negative impacts" and "significant disparities between urban and rural areas."
Response 11: The discussion was added within this recommendation.
Comment 12: Authors need to analyze the findings in this study to compare them with similar studies that have been conducted
Response 12: The discussion was added.
Comment 13: Reference number 1: is there no newer source? 1976 is too old to be used as a reference source
Response 13: Agree, a newer link has been used per your recommendation.
Comment 14: R Please correct the format for writing references number 17-21
Response 14: According to your recommendations, we have corrected the format of the links

Round 2
Reviewer 1 Report
Comments and Suggestions for Authors
author revised and gave reasonable reponse. I have no further comments